# Recycled and Nickel- or Cobalt-Doped Lead Materials from Lead Acid Battery: Voltammetric and Spectroscopic Studies

**DOI:** 10.3390/ma16134507

**Published:** 2023-06-21

**Authors:** Simona Rada, Andrei Pintea, Razvan Opre, Mihaela Unguresan, Adriana Popa

**Affiliations:** 1Physics and Chemistry Department, Technical University of Cluj-Napoca, 400020 Cluj-Napoca, Romania; 2National Institute for Research and Development of Isotopic and Molecular Technologies, 400293 Cluj-Napoca, Romania

**Keywords:** recycled electrodes from spent car battery, spectroscopic method, cyclic voltammetry

## Abstract

The active mass of the plates of aspent car battery with higher wear after an efficient desulfatization can be used as sources of a new electrode. This paper proposes the recycling of spent electrodes from a lead acid battery and the incorporation of NiO or Co_3_O_4_ contents by the melt-quenching method in order to enrich the electrochemical properties. The analysis of X-ray diffractograms indicates the gradual decrease in the sulfated crystalline phases, respectively, 4PbO·PbSO_4_ and PbO·PbSO_4_ phases, until their disappearance for higher dopant concentrations. Infrared (IR) spectra show a decreasing trend in the intensity of the bands corresponding to the sulfate ions and a conversion of [PbO_3_] pyramidal units into [PbO_4_] tetrahedral units by doping with high dopant levels, yielding to the apparition of the PbO_2_ crystalline phase. The observed electron paramagnetic resonance (EPR) spectra confirm three signals located on the gyromagnetic factor, *g*~2, 2.2 and 8 assigned to the nickel ions in higher oxidation states as well as the metallic nickel nanoparticles. This compositional evolution can be explained by considering a process of the drastic reduction in nickel ions from the superior oxidation states to metallic nickel. The linewidth and the intensity of the resonance lines situated at about *g*~2, 2.17, 4.22 and 7.8 are attributed to the Co^+2^ ions from the EPR data. The best reversibility of the cyclic voltammograms was highlighted for the samples with *x* = 10 mol% of NiO and 15 mol% of Co_3_O_4_, which are recommended as suitable in applications as new electrodes for the lead acid battery.

## 1. Introduction

Overpopulation leads to a much faster depletion of natural resources. The planet cannot produce the necessary materials by itself, and thus requires human intervention. Unfortunately, only in recent history have discussions begun with regard to how humans use natural resources at a rate that is much too high and excessively pollute the environment, thereby severely damaging the planet. This is a very important issue because, at some point, the planet will try to restore its natural balance without taking into account the wellbeing of humanity. That is why it is important to focus extensively on the processes of recycling/reusing as many materials as possible so as to utilize the resources of the planet to their full potential, resulting in a lesser impact on human beings.

As the global population grows, so does the demand for cars. This need of mankind also leads to the emergence of many wastes from this industry. An important waste is that of lead acid batteries. Such a component has existed in the construction of motor vehicles since the beginning of the internal combustion engine. In spite of the continuous development of vehicles such as by increasing speed and improving efficiency and comfort, the batteries of vehicles have not undergone major changes, as these remain to be based on the same operation principle since 1859, when they were first invented. In the construction process of a car battery, there are heavy metals, acids as well as plastic elements. These constructive elements have a strong negative impact on both human health and the environment, factors that have quite quickly made humanity aware of the danger we face and coerced into taking immediate action in connection with recycling. 

Due to various public information and battery collection campaigns, including some which refund a portion of the cost of a battery at the time of delivery to a collection point, in the United States, about 99% of the used lead from these batteries was successfully recovered.

The recycling of used electrodes from car batteries requires the integration of an eco-innovative technology, with low costs, energy efficiency and an efficient desulfatization of the plates.

The recycling methods currently applied in this field are those of pyrometallurgy and hydrometallurgy [1]. The pyrometallurgical route requires a synthesis temperature of over 1000 °C, which produces environmental problems such as the emission of sulfur oxides (70 kg/t) and lead vapors (30–50 kg/t) [2]. Hydrometallurgical processes imply the desulfurization of spent lead acid battery paste with alkaline or organic reagents [3,4].

The vacuum thermal recycling route has some advantages of the vacuum, namely: saves energy and material, prevents the contamination of gases and helps to prevent the formation of oxides [5]. 

These methods have some major drawbacks, including: requiring high efforts to revert to metal oxides, inefficient desulfatization and having a complex and toxic nature [2,3,4]. Due to these shortcomings, it was necessary to create a new eco-innovative [5,6] way of recycling, with reducing complexity and cost as the main objectives, as well as obtaining the highest possible yield (about 95%) and achieving an increased purity of the recovered substances.

Because lead has strong negative effects on humans and the environment, this new approach in battery recycling offers, in addition to the economic and energy benefits already listed, a reduction in the lead pollution emitted into the environment. This is another advantage of this method that should not be overlooked, because from the moment a quantity of lead is discharged into the environment until it affects the population, the road is short and the diseases that can be inflicted upon people are dangerous and should not be neglected.

The main inconveniences that yield to the rapid discharge of the car battery are the phenomena of anodic passivation and the evolution reactions of hydrogen and oxygen. The irreversible polarization of the anode yields passivation phenomena. This process consists of a stagnation of the metal passage—in the present case of metal lead, in the form of ions—when it plays the role of anode. Then, the presence of lead monoxide produces the premature capacity loss and the corrosion process of the positive electrode [7].

The present paper aims to recycle plates from a spent car battery and to optimize them by doping the plates with nickel (II) oxide, NiO, or cobalt (II, III) oxide, Co_3_O_4_, by a melt-quenching method; a method with a lower cost and less pollution in view of applications in the field of origin as a new electrode for the lead accumulator. Doping of the recycled material with NiO or Co_3_O_4_ was chosen because these oxides can have an effect on the reactions of hydrogen evolution. 

## 2. Experimental Procedure

The electrolyte from a spent car battery was drained, the anodic plates were separated from the cathodic ones, and the anodic plate was used as a source of lead, while the active mass from the cathode plate was useful as source of lead dioxide. 

Four host matrices based on PbO_2_–Pb in a 4:1 molar ratio were prepared at three temperatures, 950, 1000 and 1050 °C, in order to select the optimal synthesis temperature. The temperature of 1050 °C was chosen as the most suitable for the recycling of the spent electrodes due to the better homogeneity of the samples. The matrix obtained was doped with nickel oxide in the following xNiO·(100−x)[4PbO_2_·Pb] chemical formula, where *x* = 0, 1, 5, 10 and 15 mol% NiO or cobalt oxide in the xCo_3_O_4_·(100−x)[4PbO_2_·Pb] chemical formula, where *x* = 0, 1, 5, 8, 10, 15, 25 mol% Co_3_O_4_. Substances according to predetermined chemical formulas in stoichiometric proportions were weighed to the analytical balance to four decimal places (0.0001 g). Mixtures of powder substances weighed in stoichiometric proportions were ground in an agate mortar and then placed in sintered alumina crucibles. The crucibles with the weighed mixture were placed in an electric oven seated at various temperatures of: 950, 1000 or 1050 °C. After 10 min, the crucible with the melt was removed from the oven and quickly poured onto a stainless steel plate at room temperature.

The amorphous or crystalline nature of the prepared samples (the synthesized samples were finely ground into powder) was investigated by X-ray diffraction using a Shimadzu XRD-6000 diffractometer, using a graphite monochromator for a copper anode tube (with the wavelength, *λ* = 1.54 Å). IR absorption spectra were recorded at room temperature using the JASCO 6200 Fourier transform infrared (FTIR) spectrometer. EPR spectroscopy measurements were performed at room temperature, in the X frequency band, using the Adani PS 8400 spectrometer. All structural investigations, respectively, X-ray diffraction, IR and EPR spectroscopy were performed on powder samples.

The cyclic voltammetry measurements were recorded using an AUTOLAB PGSTAT 302N potentiostat/galvanostat (EcoChemie, Utrecht, The Netherlands) and NOVA 1.11 software. The samples obtained in the form of discs were used as working electrode, the platinum electrode was used as counter electrode, the calomel electrode as reference electrode and the sulfuric acid solution, H_2_SO_4_, was used as electrolyte solution. All experiments were performed in 38% H_2_SO_4_ solution to simulate the operating conditions of an electrode from the car battery.

## 3. Results and Discussions

The color of the prepared samples changes from yellow to brown by adding NiO content and from yellow to black by adding Co_3_O_4_ content, respectively, in the host matrix.

### 3.1. Structural Analysis by X-ray Diffraction (XRD)

X-ray diffractograms of the host matrix with the 4PbO_2_·Pb formula prepared at 950, 1000 and 1050 °C are illustrated in Figure 1a. For the 4PbO_2_·Pb host matrix prepared at 950 °C, four crystalline phases were found in the XRD data, namely 4PbO·PbSO_4_, Pb_2_SO_5_≡PbO⸳PbSO_4_, Pb_2_O_3_ and metallic lead crystalline phases having the main diffraction peaks centered at 2 theta: 27.62, 26.66, 29.97 and 31.3°, respectively, with 100% intensity. 

By increasing the synthesis temperature up to 1000 and 1050 °C, respectively, the peak corresponding to the metallic lead phase decreases in intensity, becoming undetectable at 1050 °C. At the synthesis temperature of 1000 °C, the 4PbO·PbSO_4_ crystalline phase predominates, and the Pb_2_SO_5_ and Pb_2_O_3_ crystalline phases are detected in the traces. At 1050 °C, the amount of Pb_2_SO_5_ crystalline phase attains maximum value, the 4PbO·PbSO_4_ crystalline phase decreases to almost undetectable amounts, and in the traces, the Pb_2_O_3_ crystalline phase is also identified.

As recycling is intended to remove the lead sulfates and oxo-sulfates content, namely 4PbO·PbSO_4_ and PbO·PbSO_4_ crystalline phases, the sample prepared at 1050 °C was selected as being suitable for doping because it has smaller amounts of the 4PbO·PbSO_4_ crystalline phase. The results suggest that the synthesis temperature has an important effect on the vitroceramic structure. For the sample prepared at 1050 °C, the amounts of oxo-sulfated crystalline phases were lowered because the lead sulfate decomposes into sulfur oxides at over 1000 °C.

X-ray diffractograms of the recycled and nickel-doped system in the xNiO·(100−x)[4PbO_2_·Pb] composition (where *x* = 0, 1, 5 and 10 mol% NiO) are represented in Figure 1b. The diffraction pattern of the sample with *x* = 0 mol% NiO confirms the presence of Pb_2_SO_5_ and Pb_2_O_3_ crystalline phases. By adding a smaller NiO content up to *x* = 1 mol%, the same crystalline phases were highlighted as in the host matrix, but there is a decrease in the intensity of diffraction peaks corresponding to the Pb_2_SO_5_ crystalline phase, which indicates that its content decreases in the vitroceramic structure.

For the samples with *x* ≥ 5 mol% NiO, new diffraction peaks characteristic of the Pb_2_O_3_, PbO_2_ and PbO crystalline phases appear. For the Pb_2_SO_5_ crystalline phase, a small diffraction peak centered at 26.91° was found, corresponding to its intensity of 100%. For the sample with the *x* = 5 mol% NiO, the Pb_2_SO_5_ crystalline phase appears in traces below the detection limit of the diffractometer (≤1–2%). By increasing the NiO level, *x* ≥ 5 mol% NiO, the content of PbO_2_ crystalline phase increased.

X-ray diffractograms of the xCo_3_O_4_·(100−x)[4PbO_2_·Pb] samples (where *x* = 0, 1, 5, 8, 10, 15, 25 mol% Co_3_O_4_) are shown in Figure 1c. For smaller dopant contents, 1 ≤ *x* ≤ 10 mol% Co_3_O_4_, the presence of diffraction peaks attributed to the 4PbO·PbSO_4_,2PbO·PbSO_4_, PbO and PbO_2_ crystalline phases was confirmed. For the sample with *x* ≥ 15 mol% Co_3_O_4_, smaller diffraction peaks corresponding to the Pb_2_O_3_=PbO·PbO_2_ crystalline phase are highlighted. For the sample with the highest dopant level (*x* = 25 mol% Co_3_O_4_) in the diffractogram, respectively, only small diffraction peaks appear, corresponding to the lead oxide crystalline phase.

By the addition of a higher dopant of over *x* > 15 mol% Co_3_O_4_, the cobalt ions ”accelerate” the decomposition of sulfated and oxo-sulfated phases in the vitroceramic network and produce a predominantly amorphous structure that overlapped with small amounts of crystalline phases, respectively, with the main diffraction peak of the PbO_2_ crystalline phase situated at 2 theta 28.47° (with 100% intensity). As a result, the cobalt ions play an important role in the efficient desulfatization of spent and recycled electrodes by the melt-quenching method.

In conclusion, for an efficient desulfatization of the spent plates from the car battery, two factors must be taken into account: (i) synthesis temperature—a high temperature achieves the PbSO_4_ decomposition and (ii) a suitable dopant content produces the efficient desulfatization of the spent electrodes up to lead oxides. For the samples with *x* ≥ 8 mol% NiO, the PbO_2_ and PbO crystalline phases were detected. X-ray diffractograms indicate a sudden decrease in the sulfated phase content and an increase in new diffraction peaks corresponding to the PbO_2_ crystalline phase for the *x*> 10 mol% Co_3_O_4_ in the host matrix. This method has advantages of lower costs, causing less pollution and efficient desulfatization processes of the spent electrodes from a car battery.

### 3.2. Structural Investigations from FTIR Spectra

The FTIR spectra of the xNiO·(100−x)[4PbO_2_·Pb] vitreous system, where *x* = 0, 1, 5 and 10 mol% NiO, are presented in Figure 2a.

IR spectra indicate the regions of the bands characteristic of different structural units in the vitroceramic structure. The IR band centered at 410 cm^−1^ implies the bending vibrations of the Pb–O–Pb/O–Pb–O angles from the [PbO_4_] structural units [8,9]. For the samples with *x* = 0% prepared at 1000 °C and *x* = 5 mol% NiO prepared at 1050 °C, this band has the maximum intensity. These structural evolutions indicate a higher content of [PbO_4_] pyramidal units in these vitroceramic structures.

The IR band centered at 600 cm^−1^ derives from the overlaps of some contributions: Pb–O stretching vibrations in the [PbO_n_] structural units, and the S–O and Ni–O stretching vibrations [10]. By increasing the dopant level, a decrease in this IR band was observed, suggesting the decrease in the sulfate ions content in the host matrix. A decrease in sulfate content is possible during synthesis because it is known that lead sulfate decomposes at temperatures above 1000 °C. The IR band reaches its maximum value for the sample with *x* = 0 mol% NiO prepared at 950 °C because it has two major oxo-sulfates phases, namely 4PbO·PbSO_4_ and PbO·PbSO_4_ crystalline phases, according to XRD data (their decomposition did not take place at this synthesis temperature). 

The IR band located at 1080 cm^−1^ is associated with vibrations of the elongation of S=O bonds from the sulfate units [11]. The intensity of this IR band reaches high values for all samples with x = 0, independent of the sintering temperature, also for the sample with *x* ≤ 1 mol% NiO. For samples with *x* ≥ 5 mol% NiO, a decrease in the intensity of the IR band centered at 1050 cm^−1^ was observed, which shows that the amount of sulfate ions was lowered.

The temperature at which the recycling process takes place must be as close as possible to the point of the thermal decomposition of lead sulfate—in this case, the temperature of 1050 °C (the 4PbO·PbSO_4_ crystalline phase disappears) is more suitable than that of 1000 °C (where the process of decomposition of the 4PbO·PbSO_4_ crystalline phase begins). The increase in the dopant level over *x* ≥ 5 mol% NiO “helps” with the thermal decomposition of the oxo-sulfate phases—in the studied case of the PbO·PbSO_4_ crystalline phase, it was detected below the detection limit of the diffractometer.

The region of IR bands centered at 610, 875, 1050 and 1150 cm^−1^ are attributed to the stretching vibrations of the Pb–O bonds in the [PbO_n_] structural units with *n* = 3, 4 and 6. The intensity of these IR bands increases by the addition of higher amounts of dopant levels, which suggests the “enrichment” of vitroceramics in crystalline phases based on lead oxides, according to XRD data. By doping the intensity of the IR band centered at 875 cm^−1^, the intensity increases, suggesting the formation of [PbO_6_] structural units. For samples with *x* ≥ 5 mol% NiO, the position of the IR band centered at 1150 cm^−1^ shifts to higher wave numbers, which indicates conversions to the [PbO_n_] structural units with *n* = 3 and 4 (predominantly *n* = 3). By doping with high levels of NiO, the transformation of the sulfated vitroceramic into a vitroceramic consisting mainly of [PbO_n_] structural units with *n* = 3, 4 and 6 (in which *n* = 3 and 6 predominate) was demonstrated.

In conclusion, the analysis of IR data indicates some structural changes depending on the synthesis temperature and the NiO concentration. If the method of obtaining the glass ceramic allows for a wider range of synthesis ±200 °C, the sintering temperature which is optimum for the recycling will be chosen as close as possible to the decomposition point of lead sulfate—in the studied case: 1050 °C. By adding a suitable NiO content, *x* > 5 mol% NiO, in the vitroceramic structure, a process of the total decomposition of lead sulfate takes place, and as a result, the formation of a mixture of lead oxides, respectively, Pb_2_O_3_, PbO_2_ and PbO crystalline phases, were evidenced. For samples with *x* > 5% mol NiO, the intensity and position of the bands corresponding to [PbO_n_] structural units with *n* = 3 and 6 were modified. For samples with *x* < 5 mol% NiO, a characteristic feature of sulfate units is highlighted at 1050 cm^−1^. For host matrices prepared at temperatures below 1050 °C, the intensity of this IR band is high. This evolution suggests the presence of a higher content of sulfated crystalline phases in host vitroceramics, according to XRD data.

FTIR spectra of the xCo_3_O_4_·(100−x)[4PbO_2_·Pb] recycled system, *x* = 0, 1, 5, 8, 10, 15, 25 mol% Co_3_O_4_ are given in Figure 2b.

(i)The first region of IR bands located in the range between 360 and 550 cm^−1^ were increased by doping. The IR band centered at 470 cm^−1^ reached its maximum value for the sample with *x* = 25 mol%, which shows the formation of the PbO_2_ crystalline phase in the vitroceramic structure, in agreement with the XRD data.(ii)The second region of IR bands centered at 600 cm^−1^ corresponds to some contributions: the stretching vibrations of the Pb–O bonds in the [PbO_n_] structural units superimposed with the Co–O and S–O bonds in the sulfate ions. At higher dopant contents, the intensity of this band decreased due to the disappearance of the oxo-sulfated phases from the vitroceramic network.(iii)The third region of IR bands located between 650 and 950 cm^−1^ is associated with the stretching vibrations of Pb–O bonds in [PbO_n_] structural units with *n* = 3, 4 and 6. The IR band centered at 875 cm^−1^ due to the [PbO_6_] octahedral units increased by doping with higher Co_3_O_4_ levels. This evolution suggests that the excess of oxygen atoms can be accommodated in the host matrix by the conversion of [PbO_3_] trigonal units into [PbO_n_] polyhedral units with *n* = 4 and 6.(iv)The last region of high-intensity IR bands situated between 950 and 1200 cm^−1^ is due to the vibrations of the S=O bonds in the sulfate units [12] (IR band located at 1080 cm^−1^) overlapping with the vibrations of the Pb–O bond in the [PbO_n_] structural units with *n* = 3 and 4. For undoped samples, the intensity of the band centered at 1080 cm^−1^ suddenly decreased as the sintering temperature increased from 850 to 1050 °C, suggesting the conversion of a significant amount of lead oxo-sulfate phases, according to XRD data.

For samples with *x* > 15 mol% Co_3_O_4_, these IR bands shift toward smaller wave numbers due to the [PbO_3_] → [PbO_4_] conversion and the formation of the PbO_2_ crystalline phase. A suitable Co_3_O_4_ content in the structure of the glass ceramic, namely *x* > 15 mol% Co_3_O_4_, produces a process of the total decomposition of lead oxo-sulfate phases and as result, the formation of the PbO_2_ crystalline phase was detected.

### 3.3. Structural Investigation by EPR Data

Electron paramagnetic resonance (EPR) data will be used to examine the coordination geometry of the nickel or cobalt ions in the vitroceramic network. 

Figure 3a shows the EPR spectra of the xNiO·(100−x)[4PbO_2_·Pb] samples where *x* = 1, 5 and 10 mol% NiO. The EPR data indicate resonance absorptions localized at *g*~2, 2.2 and 8 with a compositional dependence of their intensities.

The Ni^+2^ ions in the oxidic glasses [13] have an asymmetric resonance line normally located at *g*~2.1–2.38 [14,15,16,17,18]. The EPR signal was reported at *g*~2.7 for the lithium–bismuth–borate glasses [16]. These wide deviations of the value of g from the normal ones can occur due to: (i) the spin–orbit coupling process and the influence of the Ni^+2^ microvicinity and (ii) the super-exchange interactions between nickel ions by nonmagnetic oxygen ions that decrease the intensity of the magnetic field and increase the g value. The Ni^+3^ ions of the oxidic compounds have contributions to the wide EPR absorption band situated at *g*~2 and 2.15 [13,14,15,18,19,20,21]. This signal is characteristic to the Ni^+3^ ions situated in the pseudo-octahedral sites with rhombic structures. The Ni^+3^ ions can appear by an oxidation process of Ni^+2^ ions to Ni^+3^ ions based on the reduction in lead ions and/or during synthesis.

The wider resonance line centered at *g*~8 is attributed to metallic nickel nanoparticles [12] that can occur as a result of the reduction in nickel ions in higher oxidation states +2 (from the raw material) and +3 (during synthesis through the oxidation process).

The intensity of the resonance lines located between *g*~2 and 2.2 decreases with increasing NiO content in the vitroceramic matrix, while the wider signal centered at about *g*~8 becomes more intense by the doping process. This shows a Ni^+2^ and/or Ni^+3^ reduction process into metallic nickel nanoparticles.

The responsible mechanisms of these structural evolutions can be summarized as follows: (i) for lower dopant contents, *x* ≤ 1 mol% NiO, the resonance line centered between *g*~2 and 2.2 becomes dominant in the EPR spectrum, which indicates the presence of Ni^+2^ and several Ni^+3^ ions obtained during synthesis by the Ni^+2^ oxidation process and(ii) for samples with 5 ≤ *x* ≥ 10 mol% NiO, the intensity of the resonance line centered at *g*~8 increases while the signals situated at *g*~2 and 2.2 decreases gradually. This suggests the Ni^+2^ and Ni^+3^
→ Ni^o^ conversion. This conversion process is more pronounced for the sample with *x* = 10 mol% NiO.

EPR absorption spectra of the xCo_3_O_4_·(100−x)[4PbO_2_·Pb] samples where *x* = 1, 5, 8, 10, 15 and 25 mol% Co_3_O_4_ are recorded in Figure 3b. EPR data exhibit four signals at *g*~2, 2.17, 4.22 and 7.8. The line at *g*~4.22 is related to the high spin of Co^+2^ ions (S = 3/2) in octahedral sites while the absorption at *g*~2.17 retaliated to the isolated Co^+2^ ions (S = 3/2), occupying tetrahedral geometries. The resonance signal located at about *g*~7.8 were attributed to the S = 3/2 state of the Co^+2^ ions [22,23,24].

For the recycled and cobalt-doped samples, the contribution of three distinct features of the EPR spectrum modifies clearly with the increase in Co_3_O_4_ content. These structural evolutions can be summarized as follows:

(i) The signal at about *g*~2, including a hyperfine structure, is assigned to the coupled or clustered Co^+2^ ions overlapped with Co^+2^ ions with low spin (S = 1/2) for octahedral symmetry. The shape and feature of the resonance signal with hyperfine structure situated at about *g*~2 are very similar for the samples with *x* = 1 and 5 mol% Co_3_O_4_. For the sample with *x* = 8 mol% Co_3_O_4_, the poor eight lines of the hyperfine structure are evidenced, the intensity of the signals located at *g*~2 and 4.22 decrease and a new line centered at about *g*~2.11 appears.

In the first stage of adding 1 mol% of Co_3_O_4_ in the host matrix, the Co^+2^ ions occupy octahedral positions with high spin (*g*~4.22) and low spin (*g*~2). By increasing the dopant level up to 15 mol% Co_3_O_4_, the feature of the signals located at about *g*~2 and 2.11 broadens, changes shapes, increases in intensity and after that, their intensities decreased for *x* = 25 mol% Co_3_O_4._ For the samples with *x* = 10 and 15 mol% Co_3_O_4_, the tetrahedral geometry of the Co^+2^ ions reaches maximum value. The accommodation with oxygen excess can be realized by the conversion of octahedral positions with high spin in tetrahedral sites with high spin. After that, the addition of oxygen ions in the host matrix by up to 25% Co_3_O_4_ yields the conversion of tetrahedral into octahedral positions of the Co^+2^ ions. The following observations were made:

(ii) The contribution of the feature located at *g*~4.22 decreases by increasing cobalt concentration. At a higher content of over 8 mol% Co_3_O_4_, this feature disappears almost completely in the EPR spectrum.

(iii) The intensity of eight lines located at about *g*~7.8 increases when the Co_3_O_4_ content increased from 1 to 25 mol%. 

By increasing the cobalt content up to 25 mol% in the host matrix, an abrupt decrease in the resonance line situated at about *g*~2.17 is evidenced, the signal located at about *g*~4.22 has a weak intensity and the resonance line located at *g*~7.8 attains its maximum value. These results can be due to the conversion of four-coordinated sites into six-coordinated sites of the Co^+2^ ions and the transformation of Co^+2^ ions into Co^+3^ ions (do not present the EPR signal) by doping with higher Co_3_O_4_ levels.

The mechanism responsible for these evolutions can be described as follows: (i) for samples with *x* ≤ 5 mol% Co_3_O_4_, the Co^+2^ ions occupy the octahedral sites; (ii) for the samples with 8 ≤ *x* ≤ 15 mol% Co_3_O_4_, a conversion process of octahedral sites in tetragonal positions of the Co^+2^ ions occurs; and (iii) for a higher doping level of up to 25 mol% Co_3_O_4_, the excess of oxygen ions can be accommodated in the vitroceramic network by the transformation of tetrahedral positions into new octahedral sites of the Co^+2^ ions (the increase in intensity of the low-field resonance line situated at about *g*~7.8 due to the high spin, S = 3/2 of the Co^+2^ ions). Quantitation of EPR spectra shows that most of the Co^+2^ ions are EPR visible at a low dopant content in the host matrix and a part of the Co^+2^ ions is oxidized to Co^+3^ ions for higher Co_3_O_4_ levels. 

## 4. Cyclic Voltammetry Measurements (CV)

In the cyclic voltammetry measurements, an electrochemical cell with three electrodes was used to simulate the behavior of the electrode in a car battery: the reference electrode—calomel; the working electrode—recycled and metal-doped sample; and the counter electrode—platinum electrode, which are all immersed in a solution of sulfuric acid with a concentration of 38 %.

Cyclic voltammograms of electrode materials in the xNiO·(100−x)[4PbO_2_·Pb] composition, where *x* = 0, 1, 5 and 10 mol% NiO, are shown in Figure 4a. Their inspection indicates that the current density increases almost 100 times for the sample with *x* = 10 mol% NiO. For the recycled and nickel-doped materials with a low NiO content of up to *x* = 1 mol%, cyclic voltammograms highlight the presence of the electrode passivation phenomenon, which consists of the sudden decrease in current intensity in the positive region of the potential—after 0 V. For the electrode material with *x* = 5 mol% NiO, a well-defined oxidation wave centered at ~1.2 V is observed in the positive region of the current density, which is due to the O_2_/H_2_O redox system and the reactions of oxygen evolution.

In the region with the positive current density for the electrode material containing *x* = 10% NiO, three main oxidation waves appear centered at −0.54 V, +0.5 V and +1.68 V, and are assigned to the HPbO_2_^−^/Pb (−0.54 V), PbO/Pb (−0.58 V), Ni_2_O_3_/Ni(OH)_2_ (+0.5 V) and PbO_2_/PbSO_4_ (+1.68 V) redox systems. In the cathode region, there are two well-defined reduction peaks centered at 0.5 V, attributed to the Ni_2_O_3_/Ni(OH)_2_ redox system and −0.354 V, respectively, attributed to the PbSO_4_/Pb redox process and a small peak located at −0.126 V corresponding to the Pb^+2^/Pb^0^ redox system. For the electrode material containing *x* = 10 mol% NiO, the current density is higher than its analogue with *x* ≤ 10 mol% NiO. 

In Figure 4b, the cyclic voltammogram scanned for three cycles of the electrode materials with *x* = 10 mol% NiO is presented. This has a high degree of irreversibility because there is a pronounced dimerization effect of sulfate ions in the S_2_O_8_^−2^/2SO_4_^−2^ redox process, which appears located at 2 V, along with a change of position of the oxidation peak attributed to the Pb^+2^/Pb^0^ redox system.

These structural developments show that redox couples do not produce completely reversible reactions. The main drawbacks that lead to the irreversibility of the cyclic voltammogram are the reactions of oxygen evolution and the dimerization of sulfate ions. The presence of the dimerization reactions of sulfate ions has the disadvantage of yielding to the changes in electrolyte concentration and battery discharge. 

Our data indicate a significantly higher electrochemical performance of recycled and doped materials with *x* = 10 mol% NiO, compared to its analogues. Doping with a suitable level of nickel oxide (II) removes the phenomenon of electrode passivation and hydrogen evolution reactions, and the intensity of the residual current increases in the range 0 and 2 V. 

Cyclic voltammograms of electrodes in the xCo_3_O_4_·(100−x)[4PbO_2_·Pb] composition with *x* = 0, 1, 5, 8, 10, 15 mol% Co_3_O_4_ are shown in Figure 4c. In the cyclic voltammograms, the reactions of hydrogen evolution are not observed, and the oxidation and reduction peaks appear better formed only for samples with a higher dopant content. For this reason, we will discuss in more detail the oxidation and reduction waves for the sample with *x* = 15% moles of Co_3_O_4_. In the region with positive current density, there are two oxidation peaks centered at: −0.25 and 0.5 V. The first peak in the anodic region corresponds to an overlap of waves from the Pb^+2^/Pb^0^ redox system (located at −0.126 V) and Co^+2^/Co^0^ redox system (centered at −0.28 V). The second peak centered at 0.5 V, extending to the region above 1.23 V (much better defined and much more intense for the material with *x* = 20 mol% Co_3_O_4_), is responsible for the increase in current density in the range between 0 and 2 V. These peaks correspond to the two redox systems, namely: Co^+3^/Co^0^(wave located at +0.33 V) and O_2_/H_2_O (wave centered at 1.23 V).

In the anodic region, the following oxidation processes can be produced: Pb^0^
→ Pb^+2^ + 2e^−^(1)
Co^0^
→ Co^+2^ + 2e^−^(2)
Co^0^
→ Co^+3^ + 3e^−^(3)
2H_2_O → O_2_^0^ + 4H^+^ + 4e^−^(4)

In the cathodic region, there is a well-defined reduction peak for the electrode material containing *x* = 15 mol% Co_3_O_4_ centered at −0.35 V, corresponding to the PbSO_4_/Pb redox system (−0.345 V) superimposed with the wave from the Pb^+2^/Pb^0^ redox system (located at −0.126 V) and Co^+2^/Co^0^ redox process (centered at −0.28 V). As a result, a new reduction process appears in the cathodic region, which can be represented as follows:PbSO_4_
→ Pb^0^ + SO_4_^−2^ − 2e^−^(5)

These evolving trends in redox processes suggest that redox couples do not produce completely reversible reactions. In order to verify this hypothesis, Figure 5b shows the cyclic voltammograms scanned after three cycles of the recycled electrode materials having the composition xCo_3_O_4_·(100−x)[4PbO_2_·Pb], where *x* = 5, 8, 10, 15 mol% Co_3_O_4_. An examination of the cyclic voltammograms scanned after three cycles indicates their irreversibility for all samples studied, except for the sample with *x* = 10 mol% Co_3_O_4_. A lower degree of irreversibility is observed for the sample with *x* = 10 mol% Co_3_O_4_, but the reduction peaks are weak and much delayed.

The main processes responsible for the irreversibility of the cyclic voltammogram are due to: (i) the presence of PbO with two types of HPbO_2_^−^/Pb (located at −0.54 V) and PbO/Pb redox processes (centered at −0.58 V); (ii) the presence of cobalt ions in the Co^+2^/Co^0^ redox system (centered at −0.28 V) with a strong effect on the irreversibility of the voltammogram in the sample with *x* = 15 mol% Co_3_O_4_; (iii) the presence of cobalt ions in the Co^+3^/Co^0^ redox system (wave located at +0.33 V) with a more intense effect on the irreversibility of the voltammogram in the sample with *x* = 8 mol% Co_3_O_4_; and (iv) the presence of the sulfate dimerization process in the S_2_O_8_^−2^/2SO_4_^−2^ redox process, which appears located at 2 V, with a pronounced effect in the samples with *x* = 5 mol% Co_3_O_4_.

In conclusion, cyclic voltammetry measurements indicate clearly superior electrochemical performances of recycled and doped materials with higher cobalt double (II and III) oxide contents compared to either undoped or those with a lower dopant content, indicating a higher degree of irreversibility of redox processes. Doping with a suitable cobalt double (II and III) oxide content removes the passivation phenomenon of the electrode and the reactions of hydrogen evolution, and the intensity of the residual current increases in the range between 0 and 2 V. The lower degree of irreversibility of the cyclic voltammogram of the sample with a higher dopant level makes it suitable as an electrode for lead acid batteries. 

Linear sweep voltammograms of prepared electrode materials having the xCo_3_O_4_·(100−x)[4PbO_2_·Pb] composition, *x* = 0, 1, 5, 8, 10, 15 mol% Co_3_O_4_ in a solution of a 38% sulfuric acid concentration after scanning a cycle and having a scanning rate of 10 mV/s are illustrated in Figure 6. A simple inspection of linear scanning voltammograms indicates that the current density is increased by doping with double cobalt oxide. Maximum current density for the first oxidation peak is obtained for samples with *x* ≥ 15 mol% Co_3_O_4_, which is why it is recommended for applications as an electrode for batteries. 

The sample with *x* = 10 mol% Co_3_O_4_, although it has a slightly lower current density than its analogues with a higher dopant content, is preferable for rechargeable batteries. Thus, the incorporation of a content of 10 mol% Co_3_O_4_ in the host matrix will produce an improvement of the reversibility of the cyclic voltammogram and an increase in the number of charge–discharge processes of the battery. The new electrodes, recycled from the anodic and cathodic plates from a car battery with a high degree of wear and sulfation were optimized with a suitable content of double cobalt oxide by the melt-quenching method for application onto batteries as anodes.

More information regarding the electrochemical performances of the samples in comparison with each other and with impurities of dopants can be evidenced from parameters corresponding to the voltammetric response. Electrochemical parameters, namely the formal potential and the maximum current density intensity are listed in Table 1. For the samples doped with lower metal oxide contents, the highest values of the formal potential and the lowest values of the current density intensity showing an increase in irreversibility of the voltammogram are obtained. The comparative results of electrochemical parameters of all samples recommend that samples are doped with 10 % NiO and 15 % Co_3_O_4_ as new electrodes because they have small formal potential values and high current density values, and as result, a smaller degree of irreversibility.

In conclusion, the paper proposes an eco-innovative and efficient method for the recycling of the spent plates from a car battery with a high degree of wear and doping them with a suitable content of metal oxides in view of using the products obtained as new types of electrodes for batteries. The recycling method is more environmentally friendly, saves energy and offers an efficient desulfatization of the spent plates from car batteries. The results of various investigative techniques demonstrate the significant improvement of the conductive and electrochemical properties of the recycled electrode material containing a higher NiO or Co_3_O_4_ concentration. These performances are due to the presence of nickel or cobalt ions in different oxidation states, which have drastic but beneficial effects in changing the structure of the recycled material. Recycled and doped electrodes with a higher content of metal oxides can be reused as new energy sources in batteries.

## 5. Conclusions

The vitreous systems in the xNiO·(100−x)[4PbO_2_·Pb] composition, where *x* = 0, 1, 5, 10 mol% NiO, and with the xCo_3_O_4_·(100−x)[4PbO_2_·Pb] composition, where *x* = 0, 1, 5, 8, 10, 15, 25 mol%Co_3_O_4_, were synthesized by the proposed recycling method. The raw materials used were: (1) as the lead source, the active mass of the anode from a spent car battery and (2) as the lead dioxide source, the active mass of the cathode and NiO or Co_3_O_4_ powders from the laboratory. 

An analysis of XRD data indicates that by doping with NiO contents, the content of the Pb_2_SO_5_ crystalline phase decreases sharply and disappears for higher dopant levels, and the amount of PbO_2_ and PbO crystalline phases increase. For samples with *x* ≤ 15 mol% Co_3_O_4_, a sudden decrease in the sulfated phase content is observed and new diffraction peaks corresponding to the PbO_2_ crystalline phase appeared. For the total desulfatization of spent plates from the car battery, a dopant content of at least 15 mol% Co_3_O_4_ must be taken into account.

The IR data show a decrease in the fractions of sulfate structural units and a conversion of [PbO_3_] → [PbO_4_] structural units by doping.

The EPR data indicate three signals at *g*~2, 2.2 and 8, assigned to the Ni^+2^, Ni^+3^ and Ni^o^. The linewidth and the intensity of resonance lines from EPR spectra depend very strongly on the Co_3_O_4_ concentration. The variation in the *g* values in the range of *g*~2, 2.17, 4.22 and 7.8 can be related to the Co^+2^ ions with varied geometry. 

Analysis of cyclic voltammograms after scanning one cycle and three cycles indicates that doping the recycled material with a suitable NiO or Co_3_O_4_ can be recommended as a new method to produce a new car battery electrode.

## Figures and Tables

**Figure 1 materials-16-04507-f001:**
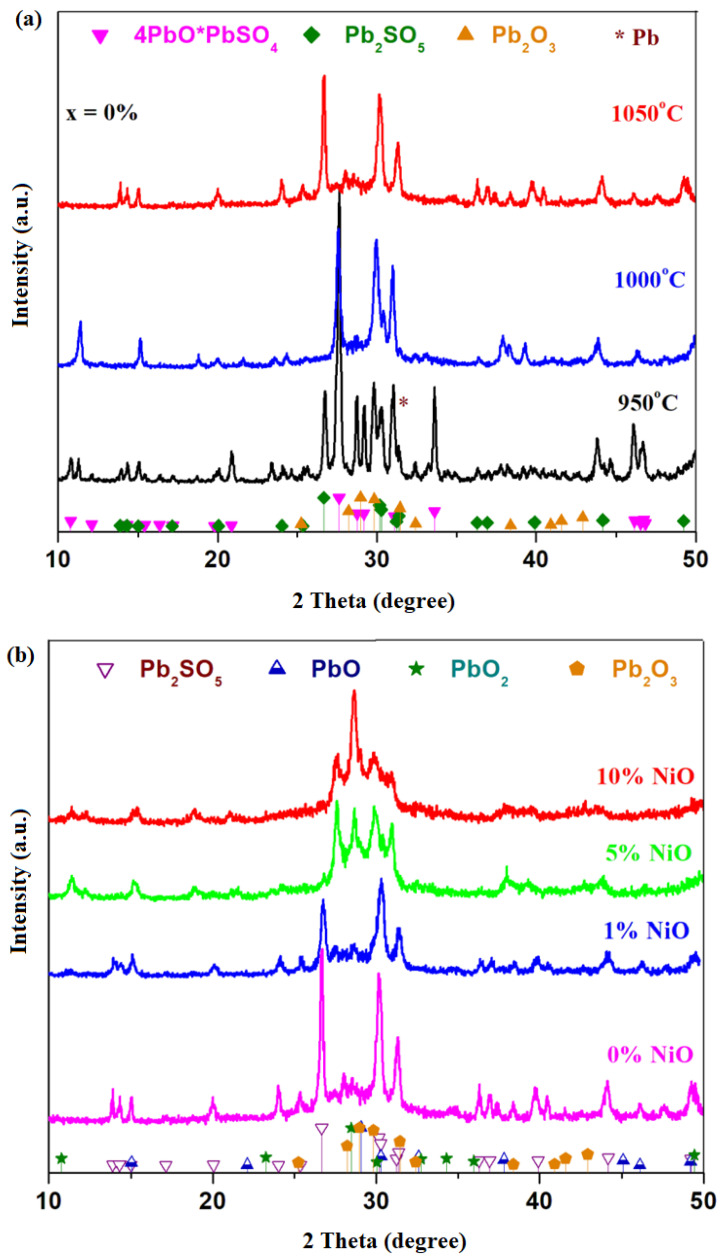
X-ray diffractograms of the recycled system and doped with nickel oxide (II) in the composition xNiO·(100−x)[4PbO_2_·Pb] where (**a**) *x* = 0 mol% NiO is synthesized at different temperatures; (**b**) *x* = 0, 1, 5 and 10 mol% NiO is prepared at 1050 °C and (**c**) in the composition xCo_3_O_4_·(100−x)[4PbO_2_·Pb] where *x* = 0–20 mol% Co_3_O_4_.

**Figure 2 materials-16-04507-f002:**
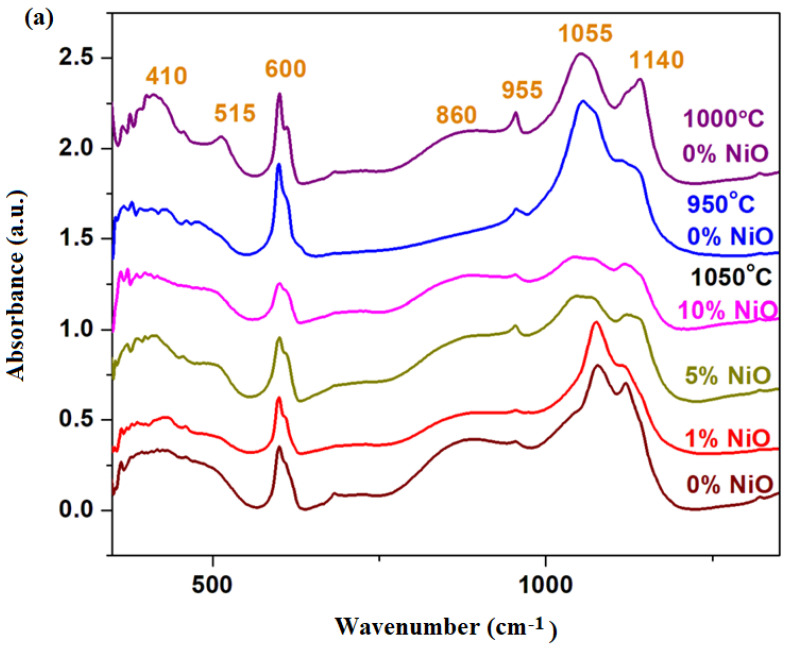
FTIR spectra of the (**a**) xNiO·(100−x)[4PbO_2_·Pb] system, where *x* = 0, 1, 5, 10 mol% NiO, prepared at 1050 °C and *x* = 0 mol% NiO (prepared at 950 and 1000 °C) and (**b**) xCo_3_O_4_·(100−x)[4PbO_2_·Pb] system, where *x* = 0, 1, 5, 8, 10, 15, 25 mol% Co_3_O_4_ (prepared at 1050 °C).

**Figure 3 materials-16-04507-f003:**
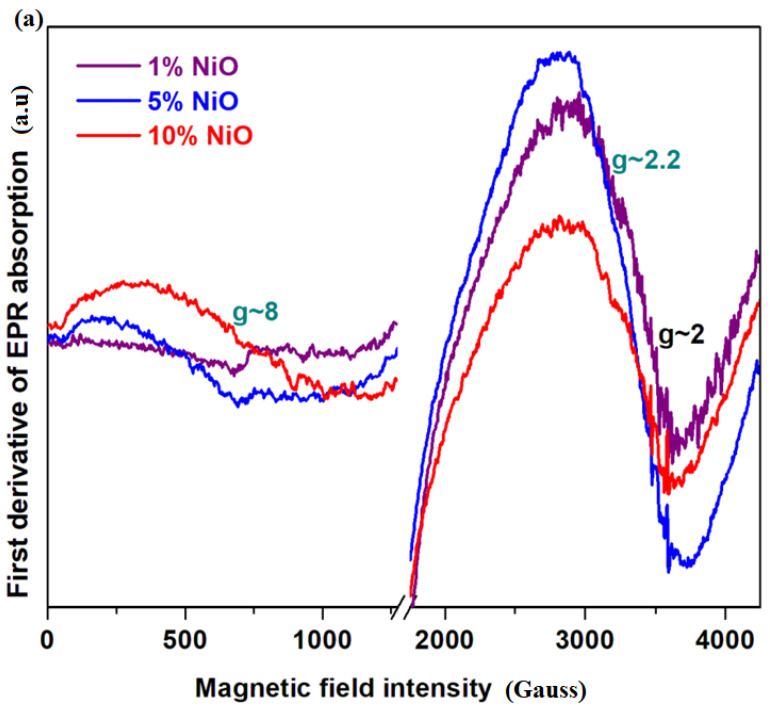
EPR spectra of (**a**) xNiO·(100−x)[4PbO_2_·Pb] system where *x* = 1, 5, 8, 10 mol% NiO. (**b**) xCo_3_O_4_·(100−x)[4PbO_2_·Pb], *x* = 1, 5, 8, 10, 15, 25 mol% Co_3_O_4_.

**Figure 4 materials-16-04507-f004:**
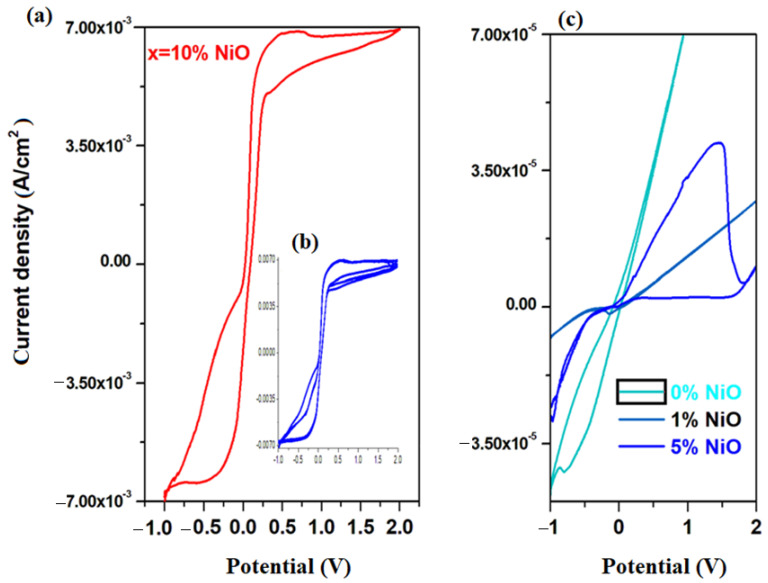
Cyclic voltammograms of the system recycled and doped with metal oxide having xNiO·(100−x)[4PbO_2_·Pb] the composition where (**a**) *x* = 10 mol% NiO and (**c**) x = 0, 1 and 5 mol% NiO prepared at 1050 °C after scanning a cycle used as electrode material in an electrochemical cell using as electrolyte a 38% solution of sulfuric acid and simulating the processes from the electrode of a car battery. (**b**) Cyclic voltammograms after scanning three cycles recorded in a solution of 38% sulfuric acid for recycled electrode materials having the composition xNiO·(100−x)[4PbO_2_·Pb] where *x* = 10 mol% NiO.

**Figure 5 materials-16-04507-f005:**
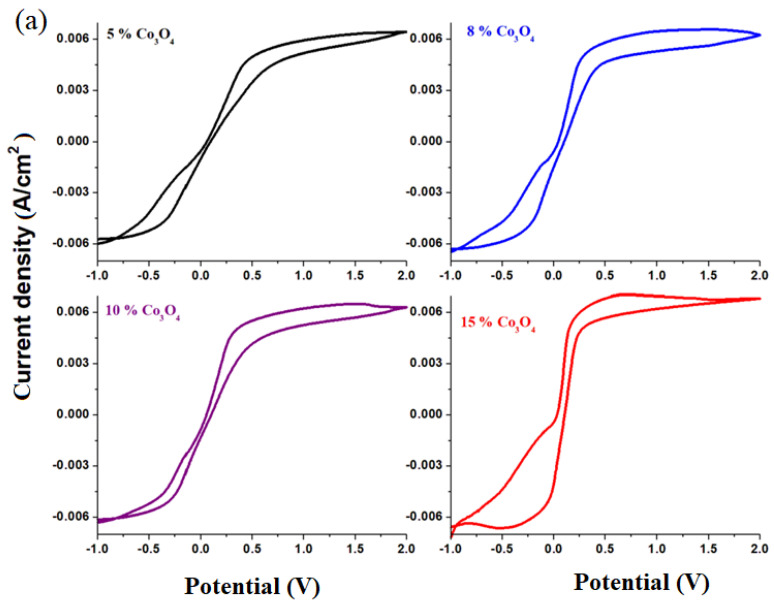
Cyclic voltammograms after scanning (**a**) a cycle and (**b**) three cycles recorded in a solution of 38% sulfuric acid for recycled electrode materials having the composition xCo_3_O_4_·(100−x)[4PbO_2_·Pb], *x* = 5, 8, 10, 15 mol% Co_3_O_4_.

**Figure 6 materials-16-04507-f006:**
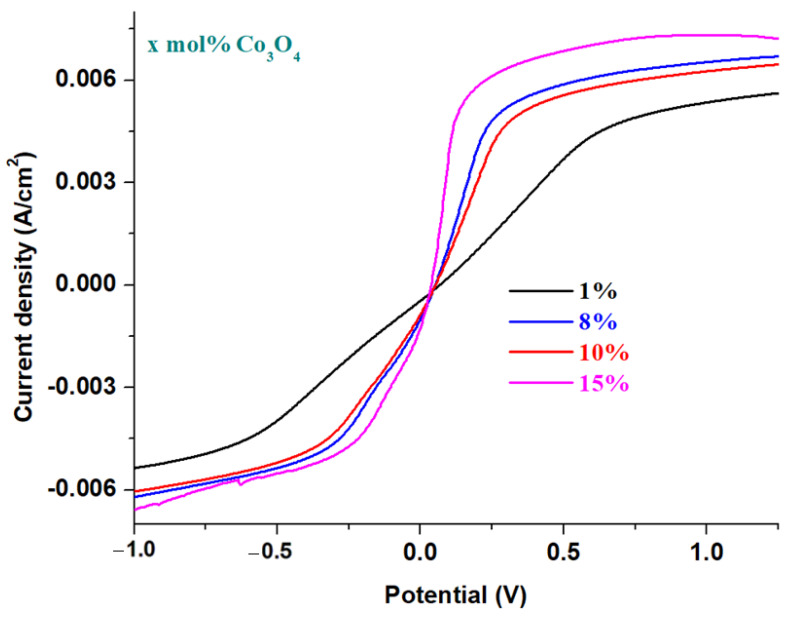
Linear scanning voltammograms recorded in 38% sulfuric acid solution with a scanning rate of 10 mV/s for prepared electrode materials having the composition xCo_3_O_4_·(100−x)[4PbO_2_·Pb] where *x* = 0, 5, 8, 10, 15 mol% Co_3_O_4_.

**Table 1 materials-16-04507-t001:** Electrochemical parameters corresponding to the linear sweep voltammetric response of the electrode material, respectively, the formal potential, E_o_,and the maximum current density intensity.

Composition of the Electrode Material	Formal Potential, E_o_ (V)	Maximum Intensity of Current Density (A/cm^2^)·10^−3^
xNiO·(100−x)[4PbO_2_·Pb]		
x = 10 mol% NiO	0.04642	6.95
xCo_3_O_4_·(100−x)[4PbO_2_·Pb]		
x = 1 mol% Co_3_O_4_	0.1536	5.37
x = 8 mol% Co_3_O_4_	0.0860	6.39
x = 10 mol% Co_3_O_4_	0.0778	6.22
x = 15 mol% Co_3_O_4_	0.0577	7.27

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
