# Peer review of "Recycled and Nickel- or Cobalt-Doped Lead Materials from Lead Acid Battery: Voltammetric and Spectroscopic Studies"

_materials, 2023, doi:10.3390/ma16134507_

Round 1

Reviewer 1 Report

This paper explored the recycling and modification of lead materials from lead-acid batteries. The topic is of interest for the lead-acid battery community. The method is simple. However, there are some issues to be addressed before consideration for acceptance.

1. The abstract is more like a conclusion. It should focus on the necessity and novelty of the paper.

2. The characterization part in the last paragraph of introduction should be removed, as it is stated in the experiment part in the following.

3. In the experiment part, 850, 900, 1000, and 1050 were chosen as the synthesis temperature. However, in the XRD analysis, the temperatures are 950, 1000, and 1050.

4. The standard peaks of various materials are mixed together, which is hard to tell.

5. In figure 6 of LSV results, it seems that the current density was not in consistency with the increase of the content of Co3O4 dopant. How to explain this?

6. The conclusion should be revised in a more concise way.

There are some typo mistakes in the manuscript. Also, many sentences are too long.

Author Response

Reviewer 1

This paper explored the recycling and modification of lead materials from lead-acid batteries. The topic is of interest for the lead-acid battery community. The method is simple. However, there are some issues to be addressed before consideration for acceptance.

  1. The abstract is more like a conclusion. It should focus on the necessity and novelty of the paper.

Authors

  1. The abstract was modified in the revised manuscript describing the necessity and novelty of this paper.

Reviewer 1

2. The characterization part in the last paragraph of introduction should be removed, as it is stated in the experiment part in the following.

Authors

  1. The last paragraph of introduction was removed in the revised manuscript.

Reviewer 1

  1. 3. In the experiment part, 850, 900, 1000, and 1050 were chosen as the synthesis temperature. However, in the XRD analysis, the temperatures are 950, 1000, and 1050.

Authors

  1. The synthesis temperatures chosen 950, 1000 and 1050 oC were also written in the experiment part.

Reviewer 1

4. The standard peaks of various materials are mixed together, which is hard to tell.

Authors

  1. The position 2theta of the main XRD peaks (of 100% intensity) was written in the XRD data.

Reviewer 1

5. In figure 6 of LSV results, it seems that the current density was not in consistency with the increase of the content of Co3O4 dopant. How to explain this?

Authors

  1. In the Table 1 was indicated the electrochemical parameters of the voltammetric response and was discussed in according with the metal oxide content level.

Reviewer 1

6. The conclusion should be revised in a more concise way.

Authors

  1. The conclusion section was rewritten shortly in the revised manuscript.

Reviewer 2 Report

In the review of research article, Recycled and nickel or cobalt-doped lead materials from lead acid battery by Rada contains so many flaws, in concepts, description and explanation. Presentation of results is good, well compiled. I suggest the authors to make the major revision.

1.       Title of the manuscript makes no sense, it’s not descriptive and informative, authors should bring some novelty in the title about their research work.

2.       English looks biggest issue with the manuscript, which must be carefully addressed dueing the revised version.

3.       What exactly is the reason of drastic reduction of Nickel in oxidation states?

4.       In the XRD analysis it is suggested to put the indexing with the intensity peaks.

Must be revised carefully by professional and technical native English speaker.

Author Response

Reviewer 2:

In the review of research article, Recycled and nickel or cobalt-doped lead materials from lead acid battery by Rada contains so many flaws, in concepts, description and explanation. Presentation of results is good, well compiled. I suggest the authors to make the major revision.

  1. Title of the manuscript makes no sense, it’s not descriptive and informative, authors should bring some novelty in the title about their research work.

  Authors:

  1. 1. The title of the revised manuscript was modified adding the work investigations.

Reviewer 1:

  1. English looks biggest issue with the manuscript, which must be carefully addressed during the revised version.

  Authors:

  1. The English language was checked in the revised manuscript.

Reviewer 2:

  1. What exactly is the reason of drastic reduction of Nickel in oxidation states?

Authors:

  1. The redox process of oxidation of the lead is responsible of drastic reduction of the nickel.

Reviewer 2:

  1. In the XRD analysis it is suggested to put the indexing with the intensity peaks.

Authors:

4. The indexing of the main intensity peaks was added in the XRD analysis.

Reviewer 3 Report

The article is devoted to the processing and alloying with nickel or cobalt of lead materials from lead acid batteries and is relevant and useful for the scientific community. However, there are a number of remarks:

1. In the introduction, it is necessary to add more citations about the problem of battery recycling and conduct a comparative analysis of effective existing methods for solving this problem.

2. Information sources must be arranged in accordance with the requirements of the publisher and citations of modern sources of information must be added.

3. In the discussion of the results, more information should be given regarding the cyclic voltammogram of the samples in comparison with each other and with impurities.

4. The conclusions should emphasize the useful scientific theoretical component of the results obtained and highlight the practical details for the technology of manufacturing electrodes for renewable batteries.

The article is devoted to the processing and alloying with nickel or cobalt of lead materials from lead acid batteries and is relevant and useful for the scientific community. However, there are a number of remarks:

1. In the introduction, it is necessary to add more citations about the problem of battery recycling and conduct a comparative analysis of effective existing methods for solving this problem.

2. Information sources must be arranged in accordance with the requirements of the publisher and citations of modern sources of information must be added.

3. In the discussion of the results, more information should be given regarding the cyclic voltammogram of the samples in comparison with each other and with impurities.

4. The conclusions should emphasize the useful scientific theoretical component of the results obtained and highlight the practical details for the technology of manufacturing electrodes for renewable batteries.

Author Response

Reviewer 3

The article is devoted to the processing and alloying with nickel or cobalt of lead materials from lead acid batteries and is relevant and useful for the scientific community. However, there are a number of remarks:

  1. In the introduction, it is necessary to add more citations about the problem of battery recycling and conduct a comparative analysis of effective existing methods for solving this problem.
  2. Information sources must be arranged in accordance with the requirements of the publisher and citations of modern sources of information must be added.

Authors

1, 2. The introduction section was improved with new citations and a comparative analysis of the existing recycling methods. The references were written in according with the journal instructions.

Reviewer 3

  1. In the discussion of the results, more information should be given regarding the cyclic voltammogram of the samples in comparison with each other and with impurities.

Authors

  1. The discussion regarding cyclic voltammograms of the samples in comparison with each other and with metallic ions was added in the revised manuscript.

Reviewer 3

  1. The conclusions should emphasize the useful scientific theoretical component of the results obtained and highlight the practical details for the technology of manufacturing electrodes for renewable batteries.

Authors

  1. The conclusion section was improved in the revised manuscript.

Round 2

Reviewer 1 Report

The authors addressed all the issues, thus it is recommended that the paper be accepted.

Reviewer 2 Report

Authors have revised the article carefully and according to my suggestions. I would like to see this article publish in the present form.

Minor spellings and grammar should be revised.

Reviewer 3 Report

The article has been finalized and can be published.